# Crosstalk between Ca^2+^ and Other Regulators Assists Plants in Responding to Abiotic Stress

**DOI:** 10.3390/plants11101351

**Published:** 2022-05-19

**Authors:** Yaoqi Li, Yinai Liu, Libo Jin, Renyi Peng

**Affiliations:** Biomedicine Collaborative Innovation Center of Zhejiang Province, Institute of Life Sciences, College of Life and Environmental Science, Wenzhou University, Wenzhou 325035, China; 20461337004@stu.wzu.edu.cn (Y.L.); 21461338012@stu.wzu.edu.cn (Y.L.)

**Keywords:** Ca^2+^, abiotic stress response, Ca^2+^ sensors, signal transduction, abiotic stress tolerance calcium, heat stress, cold stress

## Abstract

Plants have evolved many strategies for adaptation to extreme environments. Ca^2+^, acting as an important secondary messenger in plant cells, is a signaling molecule involved in plants’ response and adaptation to external stress. In plant cells, almost all kinds of abiotic stresses are able to raise cytosolic Ca^2+^ levels, and the spatiotemporal distribution of this molecule in distant cells suggests that Ca^2+^ may be a universal signal regulating different kinds of abiotic stress. Ca^2+^ is used to sense and transduce various stress signals through its downstream calcium-binding proteins, thereby inducing a series of biochemical reactions to adapt to or resist various stresses. This review summarizes the roles and molecular mechanisms of cytosolic Ca^2+^ in response to abiotic stresses such as drought, high salinity, ultraviolet light, heavy metals, waterlogging, extreme temperature and wounding. Furthermore, we focused on the crosstalk between Ca^2+^ and other signaling molecules in plants suffering from extreme environmental stress.

## 1. Introduction

Calcium ions (Ca^2+^) are important ions that maintain the normal physiological functions of plant cells and are involved in physiological metabolism in plants [1]. Ca^2+^ also functions as a ubiquitous secondary messenger involved in plant responses to various stresses [2]. Usually, there is a significant increase in the cytosolic Ca^2+^ concentration ([Ca^2+^]_cyt_) in plant cells that is caused by low temperature [3], salt [4], drought [5] and other abiotic stresses. Ca^2+^ spikes are triggered by Ca^2+^ influx through channels or Ca^2+^ efflux through pumps. This increase is recognized, amplified and transmitted downstream by Ca^2+^-binding proteins, also known as calmodulin or Ca^2+^ sensors, which regulate plant cell division, cell elongation, stomatal movement, various stress responses and growth and development through a series of conduction cascades [6].

The main function of Ca^2+^ in plant stress resistance is to stabilize plant cell walls and membranes. It can activate or inhibit various ion channels on the membrane to achieve a balance of ion concentrations inside or outside the cell. The activities of specific enzymes are activated or inhibited by Ca^2+^ in cells to regulate biochemical reactions in plants [7,8]. Moreover, the transcriptional expression of multiple anti-stress genes is regulated by changes in calcium signaling to enhance the adaptability of plants exposed to extreme environments [9]. Under abiotic stress conditions, changes in the calcium ion concentration in the plant cytoplasm can be generally recognized as a cellular secondary messenger to distinguish different original signals; it also continues to transmit the signal downstream by interacting with calcium-binding proteins, causing a series of biochemical reactions in the cells to adapt or resist various stresses [10].

Ca^2+^, considered a secondary messenger for plant signal transduction, transmits extracellular information and regulates many physiological and biochemical responses to primary signals, such as light, hormones, and gravity [11]. Cytosolic Ca^2+^ cannot be maintained at a high level for a long time. If the concentration is too high, the Ca^2+^ will react with phosphoric acid, which is necessary for the metabolism of energy substances, and produce a precipitate that inhibits the normal physiological growth of cells or even causes cell death. In normal plant cells, most Ca^2+^ exists in a bound form, collectively known as the calcium pool, and calcium-storing proteins with high capacity and low affinity for Ca^2+^ can enhance Ca^2+^-buffering capacity. Due to this low affinity, when Ca^2+^ channels in calcium banks open, Ca^2+^-binding proteins can be rapidly dissociated from Ca^2+^, releasing it into the cytoplasm so that Ca^2+^ signals can be accurately and rapidly transmitted [12]. Ca^2+^ enters the cell through the Ca^2+^ channel, which is actually a protein on the plasma membrane that is maintained in the on or off state according to changes in its conformation. This channel rapidly stimulates and induces Ca^2+^ release from the vacuole. There are two major vacuolar uptake mechanisms, including P-type Ca^2+^ pumps and a family of cation/H^+^ exchangers, which are responsible for high-affinity Ca^2+^ uptake and low-affinity with high-capacity Ca^2+^ uptake, respectively. Although research on the Ca^2+^ transport pathway mainly focuses on the regulation of [Ca^2+^]_cyt_ by calmodulin (CaM) on the cell membrane, Ca^2+^ flow through internal membrane systems, such as the endoplasmic reticulum and mitochondrial membrane, is also critical when studying the transport patterns of Ca^2+^ signals [13,14].

Because the distribution and transfer of intracellular Ca^2+^ are the basis for the formation of Ca^2+^ signals, the increase or decrease in intracellular Ca^2+^ concentrations directly affect the generation and termination of Ca^2+^ signals. When there is no external stimulation, cytosolic Ca^2+^ is insufficient to activate CaM, which lacks its own catalytic activity. However, under extreme environmental conditions, [Ca^2+^]_cyt_ increases rapidly, producing Ca^2+^ signals, and the reaction with CaM transmits the signal downward to allow subsequent physiological and biochemical reactions to occur [15]. Finally, restoration of the normal [Ca^2+^]_cyt_ levels occurs by reloading calcium stores after completing Ca^2+^ signalling, and through the calcium efflux system, which consists of Ca^2+^-ATPase pumps and Ca^2+^/H^+^ exchangers, to remove excess Ca^2+^(Figure 1) [14,16].

Calcium sensors in plants are composed of Ca^2+^-binding proteins, such as CaMs, calmodulin-like-proteins (CMLs), calcineurin-B-like proteins (CBLs), and Ca^2+^-dependent protein kinases (CDPKs). CBLs interact with CBL-interacting protein kinases (CIPKs) to form a CBL/CIPK signaling network, which plays a key role in the plant response to abiotic stress. These networks may contain many interactions, with CBLs activating CIPKs and CIPKs phosphorylating CBLs. Phosphorylation is the major mechanism affecting downstream proteins [17].

There are three major elements, influx, efflux and decoding, that affect Ca^2+^-signal translation. Ca^2+^ influx is mediated by depolarization-activated, hyperpolarization-activated and voltage-independent Ca^2+^-permeable channels, which are encoded by genes, including *cyclic nucleotide-gated channels* (*CNGCs*), *glutamate receptor-like channels* (*GLRs*), *mechanosensitive channels of small* (*MscS*) and *conductance-like channels* (*MSLs*), *annexins*, *mid1-complementing activity channels* (*MCAs*), *Piezo channels* and *channel 1* (*OSCA1*) [18]. The Ca^2+^-efflux system, the calcium-dependent protein kinase ZmCDPK7 consisting of autoinhibited Ca^2+^-ATPases (ACAs), ER-type Ca^2+^-ATPases (ECAs), and P1-ATPases (HMA1), enables Ca^2+^ efflux to form an informative signature. Specificity in Ca^2+^-based signaling is achieved via Ca^2+^ signatures with cognate Ca^2+^-binding proteins. The decoding step is carried out by protein families such as CDPKs, CBL, CIPKs, CaM and CMLs (Figure 2) [19,20].

Plants constantly suffer from various abiotic stresses during their growth and development. Ca^2+^, acting as a secondary messenger, plays an essential role in the plant response to abiotic stresses; it can not only transmit and recognize various regulatory signals but also participate in gene expression and normal protein functions [21,22]. This review summarizes the biological process of cytosolic Ca^2+^ in response to abiotic stresses, such as drought, high temperature, high salinity, heavy metals, waterlogging, and mechanical damage. Furthermore, we focus on both the crosstalk of cytosolic Ca^2+^ with other signaling molecules and biomacromolecules in plants suffering from extreme environmental stresses.

## 2. Molecular Mechanisms of Crosstalk between Ca^2+^ and Other Regulators in Response to Abiotic Stresses in Plants

### 2.1. Drought Stress

Drought is a common adverse factor inhibiting plant growth and development; high levels of drought lead to an increase in the content of reactive oxygen species (ROS) that promote membrane peroxidation and damage membrane structure [23]. Ca^2+^ plays an important regulatory role in the signaling related to the plant drought stress response, reflecting its ability to regulate the activity of some enzymes and improve the ROS-scavenging ability. In addition, damage caused by drought can be reduced with Ca^2+^ channel activation mediating stomatal closure, which reduces transpiration flux to control water loss, thus improving plant water use efficiency [24].

By monitoring the water potential of the root vascular system, plants can transmit stress signals from roots to leaves, regulate stomatal closure and induce the expression of related genes to avoid dehydration [25]. Ca^2+^ efflux was observed in epidermal cells and mesophyll cells of barley roots under drought stress conditions. Extracellular pH affects K^+^ absorption, Ca^2+^ outflow and H^+^ influx/alkalization in the leaves, which may be a chemical signal in the barley response to drought stress [26]. The application of molybdenum to wheat decreased the transpiration of wheat leaves but increased the Ca^2+^ concentration and other osmotic substances in wheat roots, which increased the osmotic pressure and further enhanced the water absorption capacity of wheat roots [27].

The abscisic acid (ABA)-dependent Ca^2+^ signaling pathway is the main response to drought stress in plants. ABA activates plasma membrane calcium channels in various ways to stimulate the release of Ca^2+^ from intracellular calcium stores, and several secondary messengers, including ROS, nitric oxide (NO), inositol 1,4,5-trisphosphate (IP3) and cyclic ADP-ribose (cADPR), are involved in this process. When water deficit occurs, ABA accumulates in the leaves. On the one hand, it activates phospholipase C and decomposes IP3, which can activate the intracellular calcium pool in guard cells to allow stomatal closure. On the other hand, intracellular Ca^2+^ can also be increased by cADPR, but no receptors for IP3 and cADPR have been identified until now in plants [28,29]. ABA can also rapidly induce an intracellular Ca^2+^ increase through hydrogen peroxide (H_2_O_2_), leading to plasma membrane hyperpolarization and direct activation of plasma membrane hyperpolarization-activated calcium channels (HACCs) and vacuolar membrane Ca^2+^ channels to achieve stomatal closure regulation [30]. At present, the pathway of NO modulating the crosstalk between ABA and H_2_O_2_ and activating the calcium signaling pathway has also been further revealed [31]. Wang et al. mentioned that extracellular Ca^2+^ and ABA promote stomatal closure by promoting H_2_O_2_ to produce calcium signals dependent on NO synthesis [32]. In *Arabidopsis* ABI mutants, H_2_O_2_ and NO activate calcium signals depending on cyclic guanosine 3′,5′-monophosphate (cGMP), which likely acts upstream of calcium signals. After exogenous calcium treatment, ion channels can be activated by intracellular calcium signaling, and calcium signal production processes mediated by ABA and H_2_O_2_ may be performed in the following sequence: ABA→H_2_O_2_→NO→cGMP→Ca^2+^. The upstream calcium-sensing signal is converted to a calcium-receiving signal, and then the downstream calcium signal can produce biological reactions promoting stomatal closure [33].

Ca^2+^ not only acts as a secondary messenger in the rapid response to upstream stimulation, but more importantly, the Ca^2+^ signaling system also contains a large number of different types of calcium signal receptors, such as CDPKs, CaM, CBL, and CIPK, which receive exogenous calcium signals and convert them into endogenous calcium signals. These signals are then phosphorylated and dephosphorylated or eventually interact with other proteins to regulate stomatal movement [34,35,36]. The interaction between CBL9 and CIPK3 negatively regulates Ca^2+^-dependent ABA signaling in *Arabidopsis* [37]. It was found that VvK1.1 in grapevine corresponds to the AKT1 channel in *Arabidopsis*, and dominates K^+^ uptake in root periphery cells. VvK1.1 and AKT1 have common functions, such as regulation by CIPK23, which occurs independently in grapevine under drought stress; this process is essential for stomatal movement regulated by K^+^ flow [38]. During stomatal closure, the relationship between ABA and Ca^2+^ is not a simple upstream and downstream regulatory process. In the early stage of drought stress, Ca^2+^ can rapidly induce ABA biosynthesis and activate Ca^2+^ channels on the plasma membrane by utilizing turgor pressure or pH change to increase the intracellular Ca^2+^ concentration instantly. Then, the expression of related transcription factors and genes, including zeaxanthin epoxidase, nine-cis-epoxy carotenoid dioxygenase, abscisic aldehyde oxidase and molybdenum cofactor sulfurase, is increased by the protein kinase cascade reaction. ABA inhibition of type 2C protein phosphatase leads to phosphorylation and activation of sucrose-nonfermenting-1-related protein kinase 2, which in turn stimulates the expression of ABA-responsive genes, thereby promoting ABA biosynthesis; then, the generated ABA in turn promotes an increase in Ca^2+^ concentration [39]. Another hypothesis is that ABA induces the activation of calcium decoding signal elements, including calcium-permeable ion channels, Ca^2+^/H^+^ antiporters and Ca^2+^-ATPases, and transduces calcium signals to alter stomatal aperture and transpiration efficiency to regulate water use efficiency in plants. Moreover, calcium channel proteins, such as *Arabidopsis thaliana two-pore channel 1* (*AtTPC1*) and *TaTPC1* from wheat, also regulate stomatal closure [40]. This hypothesis may explain the role of these genes in plant responses to drought and cold stress.

In addition to stomatal closure, plants can increase their water retention capacity by regulating stomatal density and other developmental processes to respond to drought. GT-2like 1 (GTL), a trihelix transcription family member, regulates stomatal motility by regulating the expression of *stomatal density and distribution1* (*SDD1*) genes. When PtaGTL1 identified in *Populus tremula × P. alba* was transferred to *Arabidopsis thaliana*, GTL increased *SDD1* gene expression by binding to Ca^2+^-CaM, thus reducing stomatal density and the transpiration rate and improving water use efficiency under drought stress [41]. Therefore, to adapt to different degrees of water deficit, plants adjust the stomatal number and leaf area through growth and development and balance the relationship between water use efficiency and photosynthesis to achieve the optimal adaptation point, which may be an effective strategy for plants to cope with long-term drought stress [42].

### 2.2. Salt Stress

Salt stress usually causes ion toxicity, osmotic imbalance and oxidative stress, resulting in limited plant growth and thereby affecting the sustainability of crop yields. In the external environment, hypersaline stress occurs when a high enough salt content significantly changes the water potential, thus affecting the plant [43]. Ca^2+^ also plays a significant regulatory role in plant resistance to salt stress. For example, Ca^2+^ inhibits Na^+^ influx by regulating Na^+^ entry into the main cell channel nonselective cation channels (NSCCs). Moreover, Ca^2+^ prevents the outflow of K^+^ by inhibiting K^+^ permeable outwardly rectifying conductance (KORC) channel and initiates the salt overly sensitive (SOS) signal transduction pathway, which regulates the development of plasticity in roots during salt stress adaptation; for example, SOS3 is required for auxin biosynthesis, root polar movement and the formation and maintenance of auxin gradients [44,45,46].

Usually, Ca^2+^ influx is related to hydroxyl radicals (OH·) and Ca^2+^ influx channels on the plasma membrane in wheat roots. Salt-stress-induced nicotinamide adenine dinucleotide phosphate (NADPH) oxidase on the plasma membrane produces a large number of superoxide anion radicals (O_2_^−^) extracellularly during electron transfer to O_2_, which are then rapidly converted into H_2_O_2_ and OH·. Notably, both OH· and H_2_O_2_ can activate the Ca^2+^ channels to induce extracellular Ca^2+^ flow into the cells [19]. Overall, ROS have been identified as key regulators of Ca^2+^ influx.

When suffering from salt stress, roots are the sensory part of plants that initiate the response and adaptive behavior to defend against stress damage as part of first-line defense. The SOS signaling pathway is activated by the increase in Ca^2+^ in the root cytoplasm caused by salt stress, which mediates cell signal transduction by SOS3/SCABp8-SOS2-SOS1 at the cellular level [47]. In this process, SOS3 functions as a Ca^2+^-binding protein, interacts with SOS2 to form a complex and then activates downstream SOS1 through phosphorylation, thus maintaining K^+^ and Na^+^ homeostasis inside and outside the cell. Furthermore, SOS3 has also been shown to play a key role in mediating the recombination of Ca^2+^-dependent actin filaments during salt stress [48].

CDPKs are a large polygenic family whose members contain a serine/threonine protein kinase catalytic domain as an effector region and a calmodulin-like domain for binding to Ca^2+^. These proteins can directly activate and regulate target proteins when sensing Ca^2+^ signals, thus playing an essential role in a variety of physiological processes in plants. OsCPK12 has been shown to be a crucial factor in salt stress tolerance, acting as a positive regulator of stress tolerance by regulating ABA signaling and reducing ROS accumulation in rice [49]. For example, rice overexpressing OsTPC1 show enhanced tolerance to stress through positive regulation of ABA signaling and salt signaling pathways. Some researchers suggest that ABA receptors may be upstream factors that regulate intracellular Ca^2+^ levels in plants under salt stress conditions [50]. Other experiments have shown that ABA receptors may exist inside cells or outside the plasma membrane. On the surface of the plasma membrane, when ABA acts on its receptor, the activated part interacts with G protein, which binds to the plasma membrane to activate phospholipase C and stimulate the release of Ca^2+^ from the calcium pool [51].

Ca^2+^-ATPase (PCA1) has been identified as essential for the adjustment of salt tolerance in the moss *Physcomitrella patens*. PCA1 encodes a PIIB-type Ca^2+^-ATPase, which is a plant-specific Ca^2+^ pump with an N-terminal autoinhibitory calmodulin-binding domain that has been confirmed with in vivo complementation analysis of Ca^2+^ transport-deficient yeast strains. This class of Ca^2+^ pumps may trigger the initiation of stress adaptation mechanisms in Ca^2+^ signaling pathways. In contrast to the transient [Ca^2+^]_cyt_ increase caused by NaCl in the wild-type, hyperaccumulation of cytosolic Ca^2+^ in PCA1 mutants remained high and did not return to prestimulus [Ca^2+^]_cyt_ levels. Therefore, Ca^2+^ pumps contribute to the production of stress-induced Ca^2+^ signatures [52]. In addition, based on the isolation of *monocation-induced [Ca^2+^]_i_ increases 1 Arabidopsis* mutant, which affects Ca^2+^ influx under salt stress, an association between salt sensing and GIPC-gated Ca^+2+^ influx has been inferred. It has been demonstrated that Ca^2+^ channels are gated by GIPCs in plants [53].

### 2.3. Extreme Temperature Stress

#### 2.3.1. Low-Temperature Stress

A large number of free radicals are produced in plants exposed to low-temperature stress, thereby damaging the membrane system. When plants are subjected to low-temperature stress, Ca^2+^ channels are opened, and intracellular [Ca^2+^]_cyt_ increases rapidly to induce calcium signaling [54]. Finally, the process is completed after signal transfer from the extramembrane into the membrane. On the one hand, the results of Ca^2+^ treatment of tobacco seedlings subjected to low-temperature stress showed that Ca^2+^ could increase the content of intracellular bound calcium and improve the activities of catalase, superoxide dismutase (SOD), peroxidase (POD) and other antioxidant enzymes, but reduce the content of malondialdehyde [55,56]. Furthermore, the decrease in enzyme activity after Ca^2+^ treatment was lower than that after Ca^2+^-free treatment, and the membrane permeability of tobacco seedlings also recovered quickly after growth had stopped. Therefore, it is speculated that Ca^2+^ can improve plant cold resistance and maintain the stability of the membrane system [57,58]. Another study demonstrated that Ca^2+^ and CaM could regulate the freezing resistance of citrus protoplasts, while treatment with the exogenous CaM blocker TFP or the Ca^2+^-chelating agent ethylene glycol diethyl ether diamine tetra-acetic acid (EGTA) could also inhibit the freezing resistance of citrus [59]. CBLs are a special class of Ca^2+^ receptors that specifically interact with CIPK protein kinases to activate downstream target proteins and decode Ca^2+^ signals. The expression of CIPK7 is induced by low temperature, can interact with the CBL1 protein in vitro and may be associated with CBL1 protein in vivo. Compared with wild-type plants, CBL1 mutant plants showed CIPK7 expression is affected by CBL1, suggesting that CIPK7 may bind to the calcium receptor CBL and participate in plants’ cold response [60,61].

In contrast to CaM and CBL, which have to couple with Ca^2+^ to change their conformation and be activated, CDPKs, which are constitutively activated and directly phosphorylated, transduce calcium signals by interacting with the site of the calcium receptor or forming a peptide chain [62]. CDPKs are involved in the intermediate process instead of participating in the initial response to low temperature in rice. Moreover, several Ca^2+^-related genes, such as CDPK13, are regulated by low-temperature stress in plants [63]. In rice, the CDPK13 gene is expressed in leaf sheaths and calli during the initial 2 weeks of growth, and CDPK13 is phosphorylated in response to low temperature and gibberellin (GA) signaling. Simultaneously, low temperature or exogenous GA3 treatment resulted in the elevation of CDPK13 gene expression and protein accumulation. Compared to wild-type and cold-sensitive rice, CDPK13-overexpressing-line rice showed stronger cold tolerance and a higher rate of plant recovery from cold injury, implying that CDPK13 might be a key protein in the rice signaling network responding to low-temperature stress [64,65].

Calcium channels are not only the key to the generation of calcium signals but also the rapid transport pathway and regulatory element for Ca^2+^ across the membrane [66]. At present, *Arabidopsis thaliana* two-pore channel 1 (AtTPC1) is the most studied calcium channel protein. Stomatal closure of *attpc1-2* functional deficient mutants treated with ABA, methyl jasmonate (MeJA) and Ca^2+^ was detected, and the results demonstrated that both ABA and MeJA can induce the accumulation of ROS and NO to cause an increase in [Ca^2+^]_cyt_ and cytoplasmic alkalization and activate anion channels in both wild-type and mutant plants, thus causing the stomata to be closed. However, compared with that in wild-type *Arabidopsis*, exogenous Ca^2+^ could not induce stomatal closure or activate anion channels on the plasma membrane in *attpc1-2* mutants. Taken together, we can conclude that AtTPC1 protein is involved in both stomatal closure and plasma membrane anion channel activation and is regulated by exogenous calcium signals in guard cells; however, it is not regulated by ABA and MeJA [67]. Stomatal closure is a common adaptive response of plants to low temperature. Stomatal guard cells respond quickly to abiotic stress stimuli, such as low temperature and drought [68].

Studies in eukaryotic cells suggest the overall translation rate can be regulated by an increased AMP/ATP ratio, which leads to activation of 5′-AMP-activated protein kinase and the release of Ca^2+^ from the endoplasmic reticulum, which triggers the phosphorylation of eukaryotic extension factor 2 by its activated specific kinase eukaryotic elongation factor 2 kinase [69].

#### 2.3.2. High-Temperature Stress

High-temperature stress also gives rise to plant cell membrane damage, osmotic regulation imbalance, an accumulation of ROS, an inhibition of photosynthesis, cell aging and death, thus limiting plant distribution, growth and productivity [70]. Exogenous application of Ca^2+^ effectively improves high-temperature stress resistance in laver and tomato [71,72] and alleviates the damage caused by high-temperature stress in ornamental plants such as chrysanthemum [73]. In tomato, spraying calcium chloride on the leaf surface can increase the activities of protective enzymes and soluble protein contents in leaf intima and reduce the malonic acid content, thus enhancing high-temperature-stress adaptability [71]. Further research showed that Ca^2+^ treatment can significantly improve the net photosynthetic rate, transpiration rate and stomatal conductance of tomato leaves suffering from high-temperature stress [74]. On the other hand, significant upregulation of PhCAM1 and PhCAM2 expression is related to the change in [Ca^2+^]_cyt_ when high-temperature stress occurs, while the expression of PhCAM1 and PhCAM2 is not obviously changed after EGTA is added, implying that the Ca^2+^ signaling system and CAM play a major role in the regulation of resistance to high-temperature stress in *Pyropia haitanensis* [58,75]. Based on the above descriptions, it can be clearly seen that Ca^2+^ can not only stabilize the cell membrane structure but also prevent damage to photosynthetic organs from ROS under high-temperature stress by regulating osmotic balance and the antioxidant system. Additionally, Ca^2+^, acting as an essential signaling substance, participates in signal transduction when high-temperature stress occurs and enhances high-temperature resistance in plants [76,77].

High-temperature stress also induces heat stress transcription factor (HSP) expression, and many of these factors act as molecular chaperones to prevent protein denaturation and maintain protein homeostasis [78]. Similarly to mammalian heat shock transcription factors (HSFs), plant HSFs are released from the binding and inhibition of HSP70 and HSP90 and combine with misfolded proteins under high-temperature stress. Therefore, HSFs can be used to activate the high-temperature stress response. In contrast, high-temperature stress also activates mitogen-activated protein kinases (MAPKs) and regulates the expression of HSP genes. This may be closely related not only to changes in membrane fluidity but also to calcium signaling induced by high-temperature stress, which is especially required for HSP gene expression and high-temperature stress tolerance acquisition [79,80]. The common features between signals of low- and high-temperature stress are not limited to membrane fluidity changes, calcium signaling and MAPK activation, as they also include ROS, NO, and phospholipid signaling [81,82].

The calcium-dependent protein kinase ZmCDPK7 positively regulates heat stress tolerance in maize. ABA regulates ZmCDPK7 expression by phosphorylation of the respiratory burst oxidase homologue RBOHB in a Ca^2+^-dependent manner, thus triggering ROS accumulation, which further promotes ZmCDPK7 expression. Moreover, ZmCDPK7 plays a crucial role in maintaining protein quality and reducing heat stress damage by activating the chaperone function of sHSP17.4 through Ca^2+^-dependent phosphorylation [83].

The Ca^2+^/calmodulin-dependent phosphatase calcineurin plays a role in morphogenesis and calcium homeostasis during temperature-induced mycelium-to-yeast dimorphism of *Paracoccidioides brasiliensis*. Intracellular Ca^2+^ levels increased immediately after the onset of dimorphism. The extracellular or intracellular chelation of Ca^2+^ inhibits dimorphism, while extracellular Ca^2+^ addition accelerates dimorphism. In addition, the calcineurin inhibitor cyclosporine A disrupts intracellular Ca^2+^ homeostasis and reduces mRNA transcription of the *CCH1* gene in the Ca^2+^ channel of the yeast cell plasma membrane, effectively reducing cell growth or resulting in abnormal growth morphology *P. brasiliensis* [84].

### 2.4. Heavy-Metal Stress

Increasing the Ca^2+^ content in soil can enhance the heavy-metal tolerance of plants. The accumulation of active Al^3+^ and Mn^2+^, as well as the lack of nutrients in acidic soil, are important limiting factors for crop growth [85]. Earlier studies showed that Al^3+^ could induce Ca^2+^ loss in plants and inhibit Ca^2+^ absorption and root growth, thereby suppressing plant growth and development. However, salicylic acid (SA) can alleviate Al^3+^-induced inhibition of soybean root elongation and reduce the Al^3+^ content in plants. The plant response to Al^3+^ stress requires endogenous SA and Ca^2+^ for the transmission and amplification of the Al^3+^ stress signal, which strengthens the subsequent physiological response [86]. In addition, citric acid (CA) secreted from soybean roots can alleviate Al^3+^ toxicity. Both CA secretion and SA content changes are affected by Ca^2+^, and it has been speculated that SA and Ca^2+^ might be linked to the Al^3+^ tolerance mechanism of soybean. Moreover, both Ca^2+^ and SA can alleviate the physiological reaction of root growth inhibition caused by aluminum, promote the secretion of citric acid, improve the enzyme activities of SOD, POD, ascorbate peroxidase and other antioxidant systems, reduce the accumulation of ROS, and alleviate oxidative stress damage to improve the Al^3+^ tolerance of soybean. Additionally, SA may participate in the Al^3+^ tolerance mechanism by increasing the endogenous Ca^2+^ level [87,88]. Exogenous Ca^2+^ can increase the relative expression levels of PLC and PLD genes, indicating that Ca^2+^ has some effects on the changes in phospholipase in soybean root tip cells, which may be related to changes in microtubule structure [89].

The addition of exogenous calcium can reduce the content of heavy metal ions in plants growing in soils with excessive amounts of heavy metals, such as Cu^2+^, Cr^6+^ and Pb^2+^, and improve their ability to resist heavy metal stress [90,91]. According to research findings, when the Cu^2+^ concentration increased, the Ca^2+^ content in plant roots increased, which may be significant for improving plant resistance to Cu^2+^ stress [92]. In addition, Cr^6+^ stress activates plant endogenous hydrogen sulfide (H_2_S) synthesis and Ca^2+^ signal transduction. H_2_S and Ca^2+^ alone or in combination can significantly reduce the injury caused by Cr^6+^ stress; however, the effect is better when they are used in combination. In contrast, treatment with H_2_S synthesis inhibitors or Ca^2+^ chelating agents enhances environment-induced stress. This result suggests the synergistic effects of H_2_S and Ca^2+^ in response to Cr^6+^ stress in *Setaria italica* [93].

### 2.5. Wound Stress

Usually, wounds from mechanical damage caused by harsh weather conditions, such as wind and rain, or by geological disasters, including debris flows and landslides, induce the release of calcium signals to regulate the overall response to stress and further improve the survival ability of plants [94,95]. Wound signaling is required for initiating plant regeneration. Plants promote changes in downstream cell fate due to signal transduction cascades induced by wounds [96]. Wounds also promote changes in cell membrane potential (Vm), fluctuations in Ca^2+^ concentration, ROS bursts, and drastic increases in the concentrations of jasmine, ethylene, SA and other plant hormones [97]. Therefore, Ca^2+^, as a vital part of wound signaling, may regulate the transcription of downstream genes accompanied by signal transduction and trigger some physiological and biochemical reactions locally or systemically. Studies have shown that the loss of cell membrane integrity at the site of injury may allow cytoplasmic inclusions of damaged cells to enter the intercellular space, thus changing the original ion concentration and composition, which further affects the state of various ion channels on the cell membrane and leads to fluctuations in transmembrane potential and calcium concentration [98,99]. Furthermore, GdCl_3_, a calcium channel inhibitor, has been shown to inhibit plasma membrane depolarization induced by single-cell injury [100].

Ca^2+^ signals respond to wounds rapidly (often within just 2 s) in plants suffering from mechanical damage and then propagate to specific undamaged distal tissues after 2 min. The ethylene synthesis-related genes ACS2, ACS6, ACS7 and ACS8 were rapidly upregulated within 30 min after leaf injury in *Arabidopsis.* At the same time, wounding rapidly activated the expression of mitogen-activated protein kinase (MPK) along with calcium-dependent protein kinase (CPK) [101]. To some extent, its transmission depends on glutamate receptor-like 3.3/3.6 (glR3.3/3.6) proteins, which are regulated by glutamate concentration. Mutation of both glr3.3 and glr3.6 leads to the long-distance transport of Ca^2+^ being blocked, and the expression of defense genes is subsequently reduced in undamaged areas, while glutamate contents are reduced concurrently [102]. Moreover, Ca^2+^ also functions as an intracellular secondary messenger to regulate the biochemical state of cells near wounds, and Ca^2+^-dependent MC4 in the cytoplasm has catalytic activity due to the wound-induced [Ca^2+^]_cyt_ increase. The defense response occurs by catalyzing the elicitor peptide precursors into mature peptides located on the cytoplasmic side of the vacuole membrane; in turn, these peptides are recognized by the cytoplasmic vacuolar membrane-targeted receptor-elicitor peptide receptors [103,104]. Although Ca^2+^ transfer over long distances depends on ROS produced by NADPH oxidase, inhibition of calcium ion signaling can weaken the wound response to jasmonic acid (JA) and ethylene production [97,105]. Therefore, these results indicate that there is a closely linked interaction among various substances related to wounding signals.

Ca^2+^ is directly involved in the generation and propagation of long-distance signals in plants. Under strong local stress, variation potential (VP), a long-distance intercellular electrical signal, is the potential mechanism for coordinating functional responses to different plant cells, which can cause functional changes in unstimulated organs and tissues, namely, systematic responses of plants. Specifically, ligand-dependent or mechanically sensitive Ca^2+^ channels are activated by the propagation of chemical or hydraulic signals or a combination of these potentially distant signals. Subsequent Ca^2+^ influx can trigger VP production, thus inducing H^+^-ATPase inactivation and possibly Cl^−^ channel activation [106].

In long-distance ROS signal transduction, RESPIRATORY BURST HOMOLOG D (RBOHD) is a ferric oxidoreductase that can be activated directly by calcium ions binding to its EF-Hand motif and phosphorylated by various protein kinases, such as CPK5 and CIPK. *Botrytis*-induced kinase 1 (BIK1) is also under Ca^2+^-dependent regulation by CPK28 and phosphorylates RBOHD. ROS-activated Ca^2+^-permeable channels on the plasma membrane provide a mechanism for RBOHD to trigger its further activation [107]. The crosstalk between Ca^2+^ and ROS to transmit these signals among cells across long distances, namely, that of RBOH, is activated by Ca^2+^-dependent protein kinases in the presence of Ca^2+^. This leads to the accumulation of nonprotoplast ROS, leading to induced Ca^2+^ release from adjacent cells. Then, another Ca^2+^-dependent protein kinase is activated circularly [99,108]. In this way, signals are transmitted over long distances within plants (Figure 3) [109].

Generally, Jasmonate-associated VQ domain protein 1 (JAV1) associates with JASMONATE ZIM domain protein 8 (JAZ8) and WRKY51 to form the JAV1-JAZ8-WRKY51 (JJW) complex, which inhibits the expression of jasmonate (JA) synthesis genes. Once the plant sustains an injury, the sudden increase in the concentration of Ca^2+^ causes calmodulin to sense Ca^2+^ and combine with JAV1, thus phosphorylating JAV1, depolymerizing the JJW complex, and releasing the transcriptional inhibition of the JA synthesis gene lipoxygenase 2 (LOX2), which finally results in the accumulation of large quantities of jasmine in response to wound stress [110,111].

### 2.6. Waterlogging Stress

Plants will be damaged by a lack of sufficient oxygen (O_2_) for respiration when they are exposed to waterlogging or submergence stress [112]. Under flooding conditions, when O_2_ is lacking, it will likely cause a massive buildup of CO_2_ as respiration and metabolism proceed, and when this occurs, intracellular Ca^2+^ in plants is required for the response to waterlogging-induced hypoxia stress in nonphotosynthetic organs [113,114]. Hypoxia promotes a real-time [Ca^2+^]_cyt_ increase and ROS accumulation, which may be interdependent [115]. For example, ROS in guard cells and root cells can activate Ca^2+^ channels, and Ca^2+^ can also promote ROS accumulation. Furthermore, in mutants with a loss of function of the PM-NAD(P)H oxidase subunits, ROS produced by the defective enzyme can activate Ca^2+^ channels on the cell membrane to achieve Ca^2+^ flow, thus contributing to the promotion of root tip growth. It has been demonstrated that ROS accumulation is coupled with Ca^2+^ dynamics in pollen tubes and root tips; however, relevant and reliable biochemical evidence about whether ROS directly activate NADPH oxidase is necessary [116].

Previous research showed that [Ca^2+^]_cyt_ acts as a key transducer of hypoxic signals in rice and wheat protoplasm exposed to hypoxia stress [117], and alcohol dehydrogenases (ADH), whose activity is involved in resistance to waterlogging, displayed significant improvement in maize [118]. In corn cells, hypoxic signaling rapidly elevates [Ca^2+^]_cyt_ by the release of intracellular stores of Ca^2+^; however, Ca^2+^ is not only involved in hypoxic signal transduction but also affects the activity of related Ca^2+^-dependent enzymes, such as alcohol dehydrogenase, reflecting tolerance to hypoxia [119]. Studies indicate that Ca^2+^ influx can promote the reduction in H_2_S in plants suffering from waterlogging-induced hypoxia stress [120]. H_2_S production by CBS is 3.5 times higher in the presence of Ca^2+^/CaM than in the absence of Ca^2+^, but it is inhibited by treatment with CaM inhibitors. The application of exogenous Ca^2+^ and its ion carrier A23187 markedly increased H_2_S-induced antioxidant activity, while the calcium-chelating agent EGTA, the plasma membrane channel blocker La^3+^, and calmodulin antagonists attenuated this resistance [58]. During waterlogging, hypoxia stress causes the accumulation of H_2_O_2_, activates the ROS-induced Ca^2+^ channel and triggers the self-amplifying “ROS-Ca^2+^ hub”, which further increases K^+^ loss and cell inactivation. The increased content of gamma-aminobutyric acid (GABA) induced by hypoxia is beneficial to the recovery of membrane potential and the maintenance of homeostasis between cytosolic K^+^ and Ca^2+^ signaling. In addition, the ROS-Ca^2+^ hub can be better regulated by elevated GABA through transcriptional control of RBOH gene expression, thus preventing the excessive accumulation of H_2_O_2_ and allowing plants to more easily survive waterlogging [121].

### 2.7. UV-B Radiation Stress

UV-B radiation stress not only has adverse effects on plant morphology, such as plant dwarfing and leaf thickening, but also harms plant physiological processes, including chloroplast structure damage, photosynthetic rate decreases, and transpiration weakening [122,123,124]. Studies suggest that there are at least two pathways involved in the cytoplasmic Ca^2+^ response to UV-B radiation stress in plants. On the one hand, enhanced UV-B radiation triggers a significant increase in the free Ca^2+^ concentration in the cytoplasm of wheat mesophyll cells, which may release Ca^2+^ from the intracellular calcium pool or increase intracellular Ca^2+^ influx. UV-B radiation inhibits CaM, leading to it dissociating from the inhibitory region and in turn binding to the active site, which leaves the Ca^2+^ pump in a resting state [125]. On the other hand, UV-B radiation possibly promotes phosphatase dephosphorylation in the inhibitory region and combines with the active site to play an inhibitory role [126]. In addition, the calcium pump is directly activated to change the transport of intracellular Ca^2+^ under UV-B radiation conditions, thereby increasing [Ca^2+^]_cyt_. Furthermore, a slightly increased [Ca^2+^]_cyt_ can not only act on the membrane skeleton and significantly reduce the deformability of cells but is also involved in the lipid redistribution of the membrane and the decline in membrane stability [127].

The total phenol content of wheat under UV-B+CaCl_2_ treatment increased by 10.3% compared with UV-B treatment alone. Most of the genes related to phenolic biosynthesis were upregulated during wheat germination, suggesting that exogenous Ca^2+^ promotes the accumulation of free phenols and bound phenols in germinal wheat exposed to UV-B radiation. In addition, treatment with Ca^2+^ can significantly alleviate membrane lipid peroxidation, activate antioxidant enzymes and regulate plant hormone levels. However, the Ca^2+^ channel blocker LaCl_3_ significantly reduced TPC and APX activity [128]. These contrasting results suggested that Ca^2+^ was involved in the regulation of phenolic metabolism, antioxidant enzyme activity and endogenous plant hormone levels of germinal wheat in response to UV-B radiation stress [129].

SA is considered to be a synergist of H_2_O_2_, which may contribute to the generation or maintenance of ROS signaling levels and participate in many signaling responses to abiotic stresses, such as UV-B [130] and heavy metals [86]. Ca^2+^ is essential for H_2_O_2_- and SA-mediated signal transduction. *Arabidopsis thaliana* BTB and TAZ domain proteins (AtBTs) are Ca^2+^-dependent CaM-binding proteins. The AtBT family may be a signal transduction center, and the signal transduction chain includes Ca^2+^, H_2_O_2_ and SA. These signals may regulate transcription by altering AtBT expression and conformation [131].

## 3. Calcium Ion Downstream Signaling Response

Under abiotic stress conditions, plants transmit information through a second messenger, allowing cells to transmit external information into the cell interior. The cells then respond by triggering downstream reactions, consisting of transcriptional regulation and protein modification, to influence appropriate adaptive responses [132]. For example, in response to heat stress, altered membrane fluidity is sensed through Ca^2+^ channels and receptor-like kinases. Heat stress transcription factor A1 (HsfA1) transcription factors are the main heat-stress-resistance regulatory factors in plants. When activated by heat, they target downstream transcription factors, microRNAs and *ONSEN* (a copia-like retrotransposon) to induce the expression of heat stress-responsive genes that are critical for ROS clearance, protein homeostasis and heat stress memory [133]. Downstream events of Ca^2+^ signal transduction are mainly mediated by Ca^2+^-binding proteins. In *Arabidopsis*, membrane hyperpolarization and ROS-activated Ca^2+^-permeable channels under K^+^ deficiency result in an increase in cytoplasmic Ca^2+^, and Ca^2+^ signals are sensed by specific sensors and transmitted downstream. CBL1/CBL9 recruits the cytoplasmic kinase CIPK23 to the plasma membrane, where CIPK23 activates AKT1-mediated uptake of K^+^ through phosphorylation. [134,135].

Calcium regulates the actin cytoskeleton either directly by binding to actin-binding proteins (ABPs) and regulating their activity or indirectly through calcium-stimulated protein kinases, such as CDPKs. The oscillation of the Ca^2+^ concentration gradient in the tip region of the pollen tube affects actin dynamics, and the remodeling of the actin cytoskeleton is associated with pollen tube elongation, showing that the Ca^2+^ concentration gradient may precisely regulate actin dynamics and promote pollen tube growth [136].

## 4. Conclusions and Perspectives

As one of the most important signaling molecules in cells, the Ca^2+^ signal transduction pathway is widely involved in the regulation of growth and development, abiotic stress response and many other physiological processes. Various studies have confirmed that abiotic stresses such as drought, high salt, ultraviolet light, heavy metal, waterlogging and extreme temperature can lead to a rapid increase in intracellular Ca^2+^ via the regulation of a variety of Ca^2+^ channels and trigger the Ca^2+^ signaling process. Then, the signals are decoded by Ca^2+^ sensors, following a series of physiological reactions through appropriate transduction pathways. Ca^2+^ is involved in crosstalk between other signaling molecules and phytohormone interactions when plants suffer from abiotic stress. In general, calcium, as the central node of the regulatory network, assists other regulators in adapting to adverse abiotic stresses.

Although the many molecular mechanisms behind Ca^2+^ involvement in abiotic stress responses have been elucidated, it remains unclear how plants can accurately distinguish the types and intensities of external stimuli and thus regulate [Ca^2+^]_cyt_ in a precise and complex way so that they can respond to a series of complex upstream signals accurately and exclusively and ensure signal transduction sensitivity and specificity concurrently. Furthermore, because crosstalk between Ca^2+^ and other signaling molecules is vital for the stress response, the mechanism of stress perception and the system of signal transduction at the biological level should be investigated. Therefore, the next important task for Ca^2+^ signaling research is to determine which physiological reactions are involved in the various Ca^2+^-targeted proteins downstream of calcium signaling and which downstream molecules are regulated to affect gene expression. Moreover, with recent advances in techniques and the development of molecular biology, cell biology, genetics and other disciplines, the role of Ca^2+^ signaling will certainly be elucidated more thoroughly.

## Figures and Tables

**Figure 1 plants-11-01351-f001:**
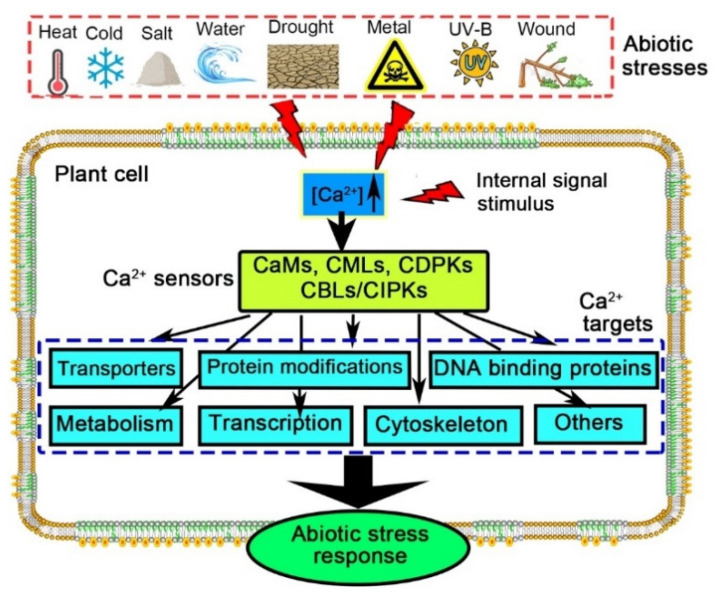
The Ca^2+^ signaling network in plant cells. Abiotic stress, including high-temperature stress (heat), low-temperature stress (cold), salt stress (salt), waterlogging stress (water), drought stress (drought), heavy-metal stress (metal), ultraviolet-B radiation stress (UV-B) and wound stress (wound), gives rise to an increase in [Ca^2+^], which is subsequently decoded by Ca^2+^ sensors such as Ca^2+^-dependent protein kinases (CDPKs), calmodulin-like-proteins (CMLs), calmodulins (CaMs), and calcineurin-B like proteins (CBLs) and their interacting protein kinases (CIPKs). These sensors activate various downstream responses that in turn result in an overall response precisely according to the original stimulus.

**Figure 2 plants-11-01351-f002:**
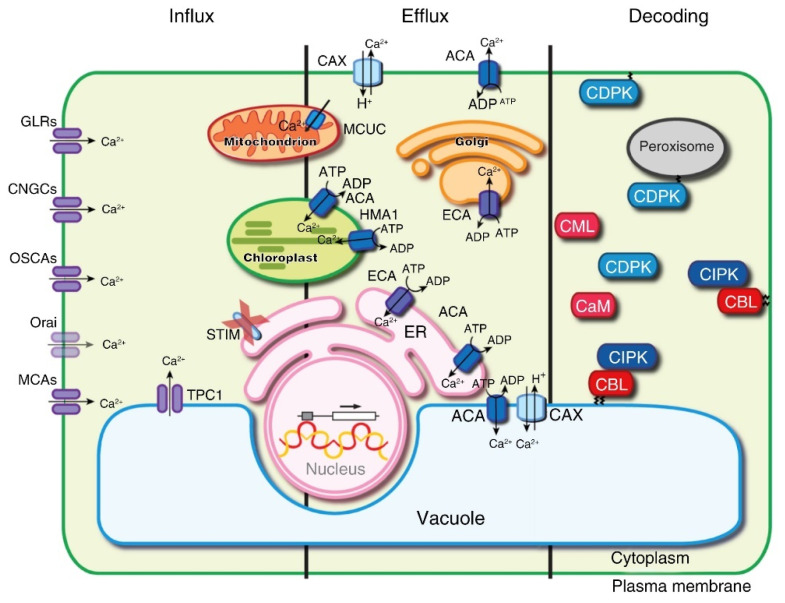
The generation and translation of Ca^2+^ signals in plant cells. Three major processes, including influx, efflux and decoding, can alter the effects of Ca^2+^-signal translation. GLRs: glutamate receptor-like channels, CNGCs: cyclic nucleotide-gated channels, OSCAs: hyperosmolality-induced Ca^2+^ increase channels, ACAs: Ca^2+^-ATPases, ECAs: Ca^2+^-ATPases, HMA1: P1-ATPases, MCUC: mitochondrial calcium uniporter complex, CAX: Ca^2+^ exchangers, CDPKs: calcium-dependent protein kinases, CBL: calcineurin B-like, and CIPKs: protein kinases. Reproduced with permission from [20], copyright 2017 Elsevier.

**Figure 3 plants-11-01351-f003:**
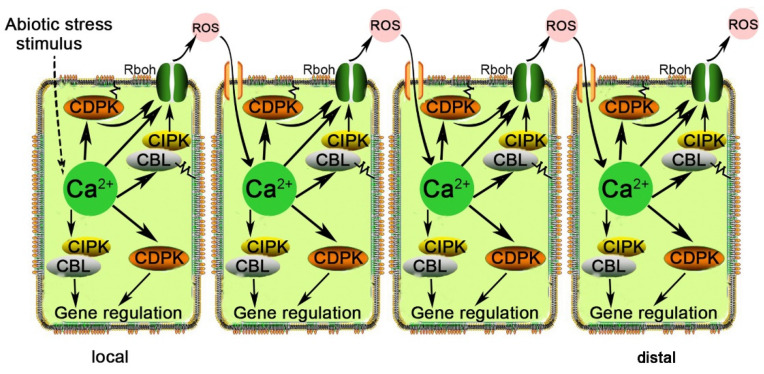
Schematic model of Ca^2+^- and ROS-mediated cell-to-cell signal propagation over long distances in plants. Stimulating the production of cytosolic Ca^2+^ signals results in the activation of RBOHD by Ca^2+^-regulated kinases, which produce ROS and then propagate the signal by activating Ca^2+^ channels in neighboring cells.

## Data Availability

Not applicable.

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
