# Peer review of "Crosstalk between Ca2+ and Other Regulators Assists Plants in Responding to Abiotic Stress"

_plants, 2022, doi:10.3390/plants11101351_

Round 1

Reviewer 1 Report

The review Crosstalk between Ca2+ and other regulators assists plants to respond to abiotic stress try to sumarize all main finding's related to the role played by calcium in different stressing conditions.

The main concern is, even as a review, it is quite superficial without adequate delving into the mechanisms of each calcium-mediate response. Besides, it would be interesting to the readers, and we believe would broaden the spectrum of potential additional readers, if some comparisons could be done for instances with other eukaryotic cells like yeasts.

On the other hand a very careful review in the English style must be done. Let's take just one example:

Lines 62-64: the authors mention the following:

Because the distribution and transfer of intracellular Ca2+ are the basis of formation of Ca2+ signal, and increase or decrease of intracellular Ca2+ concentration directly affects the generation and termination of Ca2+ signal.

Which is the meaning of this sentence? Perhaps it would be better if written like this:

Because the distribution and transfer of intracellular Ca2+ are the basis of formation of Ca2+ signal, THE increase or decrease of intracellular Ca2+ concentration directly affects the generation and termination of Ca2+ signal.

There are many other places with small mistakes!

Author Response

Response to the Reviewers' Comments

Reviewer 1:

The review Crosstalk between Ca2+ and other regulators assists plants to respond to abiotic stress try to sumarize all main finding's related to the role played by calcium in different stressing conditions.

Comment 1: The main concern is, even as a review, it is quite superficial without adequate delving into the mechanisms of each calcium-mediate response. Besides, it would be interesting to the readers, and we believe would broaden the spectrum of potential additional readers, if some comparisons could be done for instances with other eukaryotic cells like yeasts.

Response to comment 1: Thanks for reviewers’ kind advice, I'm sorry that due to the large title and limited space of this review, in order to comprehensively introduce the calcium-mediated reaction mechanism, we did our best to elaborate our views and improve and deepen them within our ability. According to your suggestion, we have added some content about eukaryotic cell “Studies in eukaryotic cells suggesting two ways to regulate the overall translation rate, increased AMP/ATP ratio leads to activation of 5'-AMP-activated protein kinase and release of Ca2+ from the endoplasmic reticulum, trigger the phosphorylation of eukaryotic extension factor 2 by its activated specific kinase eukaryotic elongation factor 2 kinase [73].” between line 273 and 276 in the revised manuscript.

And “Ca2+/calmodulin-dependent phosphatase calcineurin play a role in morphogenesis and calcium homeostasis during temperature-induced mycelium-to-yeast dimorphism of Paracoccidioides brasiliensis. Intracellular Ca2+ levels increased immediately after the onset of dimorphism. Treating with the chelation of Ca2+ with extracellular or intracellular inhibits diomorphism, while extracellular Ca2+ addition accelerates diomorphism. In addition, the effects of calcineurin inhibitor cyclosporine A disrupt intracellular Ca2+ homeostasis and reduces mRNA transcription of the CCH1 gene in the Ca2+ channel of the yeast cell plasma membrane, effectively reducing cell growth or resulting in abnormal growth morphology P. brasiliensis[87].” have been inserted between line 313 and 321 in the revised manuscript.

Comment 2: On the other hand a very careful review in the English style must be done. Let's take just one example:

Lines 62-64: the authors mention the following:

Because the distribution and transfer of intracellular Ca2+ are the basis of formation of Ca2+ signal, and increase or decrease of intracellular Ca2+ concentration directly affects the generation and termination of Ca2+ signal.

Which is the meaning of this sentence? Perhaps it would be better if written like this:

Because the distribution and transfer of intracellular Ca2+ are the basis of formation of Ca2+ signal, THE increase or decrease of intracellular Ca2+ concentration directly affects the generation and termination of Ca2+ signal.

Response to comment 2:Thanks for reviewer’s careful advice, we revised “Because the distribution and transfer of intracellular Ca2+ are the basis of formation of Ca2+ signal, and increase or decrease of intracellular Ca2+ concentration directly affects the generation and termination of Ca2+ signal.” as “Because the distribution and transfer of intracellular Ca2+ are the basis of formation of Ca2+ signal, the increase or decrease of intracellular Ca2+ concentration directly affects the generation and termination of Ca2+ signal.” in line 64 in the revised manuscript.

Comment 3: There are many other places with small mistakes!

Response to comment 3: Thanks for reviewer’s careful suggestion, I'm sorry that there are many mistakes in the review. We revised “So when there is no external stimulation, cytosolic Ca2+ is insufficient to activate CaM leading to Ca2+·CaM being in a closed state.” as “When there is no external stimulation, cytosolic Ca2+ is insufficient to activate CaM leading to Ca2+·CaM being in a closed state.” in line 65. In addition, “stress” has been deleted in line 27 in the revised manuscript.

Reviewer 2 Report

Abiotic stresses have adverse impacts on plant growth, developmental processes, crop productivity, and food quality. It is becoming clear that Ca2+ signaling plays a crucial role in plant response to various abiotic stresses. In this study, the authors focus on the molecular mechanisms of Ca2+ signaling mediated plant tolerance in abiotic stress. Meanwhile, the manuscript also stressed the importance of calcium, ROS signaling, as well as the production and the cross-talk between stress-related phytohormones such as ABA.

To sum up, the manuscript was generally well organized and well written. Furthermore, the authors present a very interesting topic. There are currently wide interests in the improvement of plant tolerance to abiotic stress. It would be of wide interest to the plant community, crop industry, and the “Plants” readers. However, I have some concerns about the manuscript, before publication:
My major concerns:
1. The authors provided four very interesting figures. Unfortunately, these figures were not fully explained in the manuscript. Moreover, figure 2 and figure 3 were not mentioned in the main text.
2. Figure legends are significant for the readers to understand the figure and manuscript. However, the authors fail to provide enough information about figure 3 and figure 4, which makes readers difficult to follow.
3. Although the title of the manuscript is “Crosstalk between Ca2+ and other regulators assists plants to respond to abiotic stress”, the authors focused on the downstream of calcium signaling in plant response to abiotic stress. In the manuscript, the authors did not mention how abiotic stress triggered Ca2+ spikes, such as the role of Ca2+ channels or pumps in abiotic stress.
My minor concerns:
1. The language in the manuscript needs to be improved.
2. In abstract line 9, “Ca2+, acting as an important second messenger in plant cells, is a sensor of plant response and adaptation to external stress.” The sentence is not accurate. Ca2+ is not the sensor of environmental stress. The plant relies largely on receptors to perceive environmental cues and subsequently activated receptor induced Ca2+ signaling. However, Ca2+ is not a direct sensor.
3. In lines 32-34, the authors need to add references related to the information.
4. In line 70, “At the same time, Ca2+ was separated from the receptor protein, namely CaM, and with signal termination, a signal transduction process is completed [21-23].” The sentence is controversial.
5. Wounding stress, especially herbivore attack, usually is considered as biotic stress. If authors still hope to keep this section, please emphasize abiotic stress.
6. In line 425, “LACL 3” should be LaCl3 (“3” should be font subscript)
7. Line 433, authors should define the meanings of “high salt”
8. About “abbreviation”, the abbreviation should be arranged based on alphabetical order. Also, please keep the definition part to align text left.

Reviewer 3 Report

The review by Li et al. is devoted to analysis of signal pathways of Ca2+ and other signaling molecules in plants. The work is interesting, however, there are some remarks:

1) Lines 45-46: words "responses to" are repeated twice.

2) Lines 73-74: salt stress is not specified.

3) Line 132: K2+ should be replaced with K+.

4) Lines 258-273: association with low temperature stress is unclear, please add an explanation.

5) P. 2.5, lines 357, 386: It is necessary to expand the description of the role of Ca2+ in the generation and propagating of long-distance signalling in plants. Maybe, you should take a look at the articles: Choi et al., 2016 (Annual Review of Plant Biology) doi: 10.1146/annurev-arplant-043015-112130

Vodeneev et al., 2015 (Plant Signaling & Behavior) doi:10.1080/15592324.2015.1057365

Thus, I suppose that revision is necessary.

Reviewer 4 Report

The goal of this review is to summarize the role of calcium in abiotic stress responses to drought, salinity, temperature stresses, waterlogging, heavy metal contamination, wounding and UV stress.  The manuscript is well organised and tries to summarise the important processes in schemes. Many references are used, but they are relevant to the text. Approximately half of them are created by reviews.

The text is mostly general with a weak description of the detailed effects of Ca. Some chapters are better written and they include a detailed summary of the Ca effects (wounding and waterlogging). However, the other chapters seem to be too general with a minimum of original research work. The conclusion is summarized in a scheme (Fig. 4), which is not well made and the arrows are not readable after printing. Moreover, the scheme contains gene names from different species which should be described in the legend as well as the abbreviations. Similarly, the Fig. 1 is too dark after printing and the frames should be lighter. Fig. 2 is missing the description of influx, which should be in detail mentioned also in the text (not only "and so on", l. 170). This issue is not discussed in the manuscript, but it is very important.

The second weakness is that a review should contain some remarks and it should highlight the missing knowledge. It should support to study of some concrete problems/questions, not only the general statements like which physiological reactions are involved in Ca2+ downstream signalling. Moreover, this topic (downstream signalling) is not well summarized in this review.

The manuscript needs English correction because some sentences are not grammatically correct or clearly written. The usage of abbreviations is not consistent. Some words are not necessary to be involved in abbreviations (like O2, Ca2+ and the other commonly known abbreviations). On the other hand, many protein abbreviations are not explained in the text - some short description and the whole name should be included (for example adding information if the protein is a channel, a kinase, a biosynthetic enzyme...). Now, it is not clear to the reader (for example lines 396-401). Moreover, it could be better to add the name of the species when some specific gene is mentioned (e.g. CIPK23 is from grapevine, l. 130). It is important because the numbers can differ between species.

I suggest adding keywords "calcium", "heat stress" and "cold stress" into the abstract or keywords in order to facilitate the finding.

I do not understand these words in the context: "cell solution" (l. 46), "binding protein" (l. 56), the regulation process described on l. 47, "has also been further revealed" (l. 113), "Wang et al. found" - it is a review, not an original article, so they did not found it (l. 113), "protein kinase cascade reaction" (l. 138 - which cascade?), "closely related" (l. 162), "influx to form information signature" (l. 172), "intracellular calcium pool" (l. 206 - really intracellular?), the sentence on l. 245, "difference in the intermediate process (l. 248), "similar to plants facing low temperature stress" (l. 278 - I do not think that; heat and cold stresses are very different in the mentioned effects), l. 299-300.

I also miss the connection of these sentences with the previous text and with the specific abiotic stress (it is mostly common): l. 337-339, l. 168-175, l. 270-273, l. 290-295. RBOH (l. 387) were for example found also in other stresses (heat stress).

Why do you miss a detailed characterization of the Ca2+ impact in roots and the water flow process in the chapter about drought? (mentioned on l. 98 only).

L. 254-257 - it was not CDPK13 mutant, but overexpressing line! It could be good to mention that the response was opposite in cold stress and in salt and drought stresses.

Correct "K2+" (l. 132), enzyme names (l. 136-143), "CDKP" to "CDPK" (repeatedly), SUMOylation is not ubiquitination and the SUMO-labelled proteins are not degraded (l. 308), GdCl3 (l. 366), Fig. 3 "fiscal".

Round 2

Reviewer 1 Report

No further comments

Author Response

Thanks for the review

Reviewer 2 Report

There are some outstanding improvements in the revised manuscript. In addition, I appreciated that the authors answered my concerns. In sum, the manuscript was organized and, generally, well written. Therefore, I suggested the manuscript is ready to be published.

Author Response

Thank you for your comments, and we will try our best to make our review meet publication requirements.

Reviewer 4 Report

I agree with all changes. The text is now readable and comprehensive.

I found only minor additional notes/mistakes:

l. 36 – add a gap „Ca2+in“

l. 153 – shortly explain the function of AKT1 (K+ channel)

l. 170 – TaTPC1 is from wheat – you should add this information or combine the text just to „such as two-pore channel 1 (TPC1),“

l. 176 – add (SDD1) after the gene name

l. 176 – PtaGTL1 was from which organism? (add it into the text)

l. 219 – you probably meant „Ca2+ efflux“ not „influx“

l. 294 – within this sentence, it is not clear which treatment is mentioned on this line in the connection with ABA as well as the cold/freezing tolerance increase is not clear

l. 326 – correct „om“

l. 352 – write the whole Latin name of the species

l. 542 – add „alcohol dehydrogenase (ADH)“

l. 605 – describe ONSEN (retrotransposon)
